# ENCODING MUSICAL STYLE
# WITH TRANSFORMER AUTOENCODERS

## ABSTRACT

We consider the problem of learning high-level controls over the global structure of sequence generation, particularly in the context of symbolic music generation with complex language models. In this work, we present the Transformer autoencoder, which aggregates encodings of the input data across time to obtain a global representation of style from a given performance. We show it is possible to combine this global embedding with other temporally distributed embeddings, enabling improved control over the separate aspects of performance style and and melody. Empirically, we demonstrate the effectiveness of our method on a variety of music generation tasks on the MAESTRO dataset and an internal dataset with 10,000+ hours of piano performances, where we achieve improvements in terms of log-likelihood and mean listening scores as compared to relevant baselines.

## 1 INTRODUCTION

There has been significant progress in generative modeling, particularly with respect to creative applications such as art and music (Oord et al., 2016; Engel et al., 2017b; Ha & Eck, 2017; Huang et al., 2019a; Payne, 2019). As the number of generative applications increase, it becomes increasingly important to consider how users can interact with such systems, particularly when the generative model functions as a tool in their creative process (Engel et al., 2017a; Gillick et al., 2019) To this end, we consider how one can learn high-level controls over the global structure of a generated sample. We focus on symbolic music generation, where Music Transformer (Huang et al., 2019b) is the current state-of-the-art in generating high-quality samples that span over a minute in length.

The challenge in controllable sequence generation is the fact that Transformers (Vaswani et al., 2017) and their variants excel as language models or in sequence-to-sequence tasks such as translation, but it is less clear as to how they can: (1) *learn* and (2) *incorporate* global conditioning information at inference time. This contrasts with traditional generative models for images such as the variational autoencoder (VAE) (Kingma & Welling, 2013) or generative adversarial network (GAN) (Goodfellow et al., 2014) which typically incorprate global conditioning as part of their training procedure (Sohn et al., 2015; Sønderby et al., 2016; Isola et al., 2017; Van den Oord et al., 2016).

In this work, we introduce the Transformer autoencoder, where we aggregate encodings across time to obtain a holistic representation of the performance style. We show that this learned global representation can be incorporated with other forms of structural conditioning in two ways. First, we show that given a performance, our model can generate performances that are similar in style to the provided input. Then, we explore different methods to combine melody and performance representations to harmonize a melody in the style of the given performance. In both cases, we show that combining both global and fine-scale encodings of the musical performance allows us to gain better control of generation, separately manipulating both the style and melody of the resulting sample.

Empirically, we evaluate our model on two datasets: the publicly-available MAESTRO (Hawthorne et al., 2019) dataset, and an internal dataset of piano performances transcribed from 10,000+ hours of audio (Anonymous for review). We find that the Transformer autoencoder is able to generate not only performances that sound similar to the input, but also accompaniments of melodies that follow a given style, as shown through both quantitative and qualitative experiments as well as a user listening study. In particular, we demonstrate that our model is capable of adapting to a particular musical style *even in the case* where we have one single input performance.

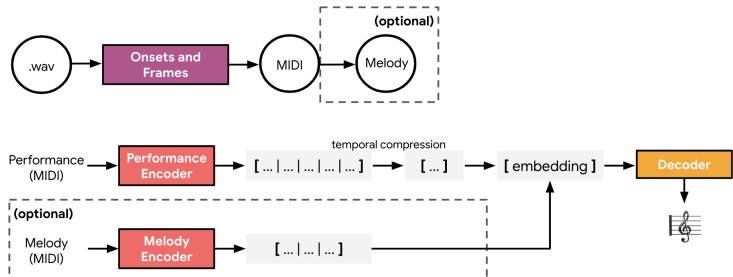

Figure 1: A flowchart of the Transformer autoencoder. We first transcribe the `.wav` data files into MIDI using the Onsets and Frames framework, then encode them into performance representations to use as input. The output of the performance encoder is aggregated across time and (optionally) combined with a melody embedding to produce a representation of the entire performance, which is then used by the Transformer decoder at inference time.

## 2 PRELIMINARIES

### 2.1 DATA REPRESENTATION FOR MUSIC GENERATION

The MAESTRO (Hawthorne et al., 2019) dataset consists of over 1,100 classical piano performances, where each piece is represented as a MIDI file. The internal performance dataset consists of over 10,000 hours of piano performances transcribed from audio (Anonymous for review). In both cases, we represent music as a sequence of discrete tokens, effectively formulating the generation task as a language modeling problem. The performances are encoded using the vocabulary as described in (Oore et al., 2018), which captures expressive dynamics and timing. This performance encoding vocabulary consists of 128 `note_on` events, 128 `note_off` events, 100 `time_shift` events representing time shifts in 10ms increments from 10ms to 1s, and 32 quantized `velocity` bins representing the velocity at which the 128 `note_on` events were played.

### 2.2 MUSIC TRANSFORMER

We build our Transformer autoencoder from Music Transformer, a state-of-the-art generative model that is capable of generating music with long-term coherence (Huang et al., 2019b). While the original Transformer uses a self-attention mechanism that operates over absolute positional encodings of each token in a given sequence (Vaswani et al., 2017), Music Transformer replaces this with *relative* attention (Shaw et al., 2018), which allows the model to keep better track of regularity based on event orderings and periodicity in the performance. Huang et al. (2019b) propose a novel algorithm for implementing relative self-attention that is significantly more memory-efficient, enabling the model to generate musical sequences over a minute in length. For more details regarding the self-attention mechanism and Transformers, we refer the reader to (Vaswani et al., 2017; Parmar et al., 2018).

## 3 CONDITIONAL GENERATION WITH THE TRANSFORMER AUTOENCODER

### 3.1 MODEL ARCHITECTURE

We leverage the standard encoder and decoder stacks of the Transformer as a foundation for our model, with minor modifications that we outline below.

**Transformer Encoder:** For both the performance and melody encoder networks, we use the Transformer's stack of 6 layers which are each comprised of a: (1) multi-head relative attention mechanism; and a (2) position-wise fully-connected feed-forward network. The performance encoder takes as input the event-based performance encoding of an input performance, while the melody encoder learns an encoding of the melody which has been extracted from the input performance. Depending on the music generation task, which we elaborate upon in Section 3.2, the encoder output(s) are fed into the Transformer decoder. Figure 1 describes the way in which the encoder and decoder networks are composed together.

**Transformer Decoder:** The decoder shares the same structure as the encoder network, but with an additional multi-head attention layer over the encoder outputs. At each step of generation, the decoder takes in the output of the encoder, as well as each new token that was previously generated.

The model is trained end-to-end with maximum likelihood. That is, for a given sequence $x$ of length $n$, we maximize $\log p_\theta(x) = \sum_{i=1}^n \log p_\theta(x_i | x_{<i})$ with respect to the model parameters $\theta$.

## 3.2 Conditioning Mechanism

**Performance Conditioning and Bottleneck**    For this task, we aim to generate samples that sound "similar" to a conditioning input performance. We incorporate a bottleneck in the output of the Transformer encoder in order to prevent the model from simply memorizing the input (Baldi, 2012). Thus, as shown in Figure 1, we mean-aggregate the performance embedding across the time dimension in order to learn a global representation of style. This mean-performance embedding is then fed into the autoregressive decoder, where the decoder attends to this global representation in order to predict the appropriate target. Although this bottleneck may be undesirable in sequence transduction tasks where the input and output sequences differ (e.g. translation), we find that it works well in our setting where we require the generated samples to be *similar in style* to the input sequence.

**Melody & Performance Conditioning:**    Next, we synthesize any given melody in the style of a *different* performance. Although the setup shares similarities to that of the melody conditioning problem in (Huang et al., 2019b), we note that we also provide a conditioning performance signal, which makes the generation task more challenging. During training, we follow an internal procedure to extract melodies from performances in the training set, quantize the melody to a 100ms grid, and encode it as a sequence of tokens that uses a different vocabulary than the performance representation. We then use two distinct Transformer encoders (each with the same architecture) as in Section 3.1 to separately encode the melody and performance inputs. The melody and performance embeddings are combined to use as input to the decoder.

We explore various ways of combining the intermediate representations: (1) `sum`, where we add the performance and melody embeddings together; (2) `concatenate`, where we concatenate the two embeddings separated with a `stop` token; and (3) `tile`, where we tile the performance embedding across every dimension of time in the melody encoding. In all three cases, we work with the mean-aggregated representation of the input performance. We find that different approaches work better than others on some dataets, a point which we elaborate upon in Section 5.

**Input Perturbation**    In order to encourage the encoded performance representations to generalize across various melodies, keys, and tempos, we draw inspiration from the denoising autoencoder (Vincent et al., 2008) as a means to regularize the model. For every target performance from which we extract the input melody, we provide the model with a *perturbed* version of the input performance as the conditioning signal. We allow this "noisy" performance to vary across two axes of variation: (1) `pitch`, where we artificially shift the overall pitch either down or up by 6 semitones; and (2) `time`, where we stretch the timing of the performance by at most 5%. In our experiments, we find that this augmentation procedure leads to samples that sound more pleasing (Oore et al., 2018). We provide further details on the augmentation procedure in Appendix A.

## 4 Similarity Evaluation on Performance Features

Although a variety of different metrics have been proposed to quantify both the quality (Engel et al., 2019) and similarity of musical performances relative to one another (Yang & Lerch, 2018; Hung et al., 2019), the development of a proper metric to measure such characteristics in music generation remains an open question. Therefore, we draw inspiration from (Yang & Lerch, 2018) to capture the style of a given performance based its the pitch- and rhythm-related features using 8 features:

1. *Note Density (ND)*: The note density refers to the average number of notes per second in a performance: a higher note density often indicates a fast-moving piece, while a lower note density correlates with softer, slower pieces. This feature is a good indicator for rhythm.
2. *Pitch Range (PR)*: The pitch range denotes the difference between the highest and lowest semitones (MIDI pitches) in a given phrase.

3. *Mean Pitch (MP) / Variation of Pitch (VP)*: Similar in vein to the pitch range (PR), the average and overall variation of pitch in a musical performance captures whether the piece is played in a higher or lower octave.

4. *Mean Velocity (MV) / Variation of Velocity (VV)*: The velocity of each note indicates how hard a key is pressed in a musical performance, and serves as a heuristic for overall volume.

5. *Mean Duration (MD) / Variation of Duration (VD)*: The duration describes for how long each note is pressed in a performance, representing articulation, dynamics, and phrasing.

## 4.1 OVERLAPPING AREA (OA) METRIC

To best capture the salient features within the periodic structure of a musical performance, we used a sliding window of 2s to construct histograms of the desired feature within each window. We found that representing each performance with such relative measurements better preserved changing dynamics and stylistic motifs across the entire performance as opposed to a single scalar value (e.g. average note density across the entire performance).

Similar to (Yang & Lerch, 2018; Hung et al., 2019), we smoothed the histograms obtained by fitting a Gaussian distribution to each feature – this allowed us to learn a compact representation while still capturing the feature's variability through its mean $\mu$ and variance $\sigma^2$. Then to compare two performances, we computed the Overlapping Area (OA) between the Gaussian pdfs of each feature to quantify their similarity. We demonstrate empirically that this metric identifies the relevant characteristics of interest in our generated performances in Section 5.

## 5 EXPERIMENTS

**Datasets** We used both MAESTRO (Hawthorne et al., 2019) and internal datasets (Simon et al., 2019) for the experimental setup. We used the standard 80/10/10 train/validation/test split from MAESTRO v1.0.0, and augmented the dataset by 10x using pitch shifts of no more than a minor third and time stretches of at most 5%. We note that this augmentation is distinct from the noise-injection procedure referenced in Section 3: the data augmentation merely increases the size of the initial dataset, while the perturbation procedure operates *only* on the input performance signal. The internal dataset did not require any additional augmentation.

**Experimental Setup** We implemented the model in the Tensor2Tensor framework (Vaswani et al., 2017), and used the default hyperparameters for training: 0.2 learning rate with 8000 warmup steps, `rsqrt_decay`, 0.2 dropout, and early stopping for GPU training. For TPU training, we use AdaFactor with the `rsqrt_decay` and learning rate warmup steps to be 10K. We adopt many of the hyperparameter configurations from (Huang et al., 2019b), where we reduce the query and key hidden size to half the hidden size, use 8 hidden layers, use 384 hidden units, and set the maximum relative distance to consider to half the training sequence length for relative global attention. We set the maximum sequence length to be 2048 tokens, and a filter size of 1024. We provide additional details on the model architectures and hyperparameter configurations in Appendix C.

## 5.1 LOG-LIKELIHOOD EVALUATION

As expected, we find that the Transformer autoencoder framework with the encoder output bottleneck outperforms other baselines. In Tables 1 and 2, we see that all conditional model variants outperform their unconditional counterparts. Interestingly, we find that for the melody & performance model, different methods of combining the embeddings work better for different datasets. For example, `concatenate` led to the lowest NLL for the internal dataset, while `sum` outperformed all other variants for MAESTRO. We report NLL values for both MAESTRO and the internal dataset for the perturbed-input model variants in Appendix B.

## 5.2 SIMILARITY EVALUATION

We use the OA metric from Section 4 to evaluate whether using a conditioning signal in both the (a) performance autoencoder and (b) melody & performance autoencoder produces samples that are more similar in style to the conditioning inputs from the evaluation set relative to other baselines.

| Model variation | MAESTRO | internal |
|---|---|---|
| Unconditional model with rel. attention (Huang et al., 2019b) | 1.840 | 1.49 |
| Performance autoencoder with rel. attention (ours) | **1.799** | **1.384** |

Table 1: Note-wise test NLL on the MAESTRO and internal datasets, with event-based representations of lengths $L = 2048$. We exclude the performance autoencoder baseline (no aggregation) as it memorized the data (NLL = 0). Conditional models outperformed their unconditional counterparts.

| Model variation | MAESTRO | internal |
|---|---|---|
| Melody-only Transformer with rel. attention (Huang et al., 2019b) | 1.786 | 1.302 |
| Melody & performance autoencoder with rel. attention, sum (ours) | **1.706** | 1.275 |
| Melody & performance autoencoder with rel. attention, concat (ours) | 1.713 | **1.237** |
| Melody & performance autoencoder with rel. attention, tile (ours) | 1.709 | 1.248 |

Table 2: Note-wise test NLL on the MAESTRO and internal datasets with melody conditioning, with event-based representations of lengths $L = 2048$. We note that `sum` worked best for MAESTRO, while `concatenate` outperformed all other baselines for the internal dataset.

First, we sample 500 examples from the evaluation set and use them as conditioning signals to generate one sample for each input. Then, we compare each conditioning signal to: (1) the generated sample and (2) an unconditional sample. We compute the similarity metric as defined in Section 4 pairwise and take the average over 500 examples. As shown in Table 3, we find that the performance autoencoder generates samples that have 48% higher similarity overall to the conditioning input as compared to the unconditional baseline.

| **MAESTRO** | ND | PR | MP | VP | MV | VV | MD | VD | Avg |
|---|---|---|---|---|---|---|---|---|---|
| Performance (ours) | **0.651** | **0.696** | **0.634** | **0.689** | **0.693** | **0.732** | **0.582** | **0.692** | **0.67** |
| Unconditional | 0.370 | 0.466 | 0.435 | 0.485 | 0.401 | 0.606 | 0.385 | 0.529 | 0.46 |
| **Internal Dataset** | | | | | | | | | |
| Performance (ours) | **0.731** | **0.837** | **0.784** | **0.838** | **0.778** | **0.835** | **0.785** | **0.827** | **0.80** |
| Unconditional | 0.466 | 0.561 | 0.556 | 0.578 | 0.405 | 0.590 | 0.521 | 0.624 | 0.54 |

Table 3: Average overlapping area (OA) similarity metrics comparing performance conditioned models with unconditional models. Unconditional and Melody-only baselines are from (Huang et al., 2019b). The metrics are described in detail in Section 4. The samples in this quantitative comparison are used for the listener study shown in the left graph of Figure 4.

For the melody & performance autoencoder, we sample 717*2 distinct performances – we reserve one set of 717 for conditioning performance styles, and the other set of 717 we use to extract melodies in order to synthesize in the style of a *different* performance. We compare the melody & performance autoencoder to 3 different baselines: (1) one that is conditioned only on the melody (Melody-only); (2) conditioned only on performance (Performance-only); and (3) an unconditional language model. Interestingly, we find that the OA metric is more sensitive to the performance style than melodic similarity. Table 4 demonstrates that the Melody-only autoencoder suffers without the performance conditioning, while the Performance-only model performs best. The Melody & performance autoencoder performs comparably to the best model.

## 5.3 INTERPOLATIONS

### 5.3.1 PERFORMANCE AUTOENCODER

In this experiment, we test whether the performance autoencoder can successfully interpolate between different performances. First, we sample 1000 performances from the internal test set (100 for MAESTRO, due to its smaller size), and split this dataset in half. The first half we reserve for

| MAESTRO | ND | PR | MP | VP | MV | VV | MD | VD | Avg |
|---|---|---|---|---|---|---|---|---|---|
| Melody & perf. (ours) | **0.650** | **0.696** | 0.634 | 0.689 | **0.692** | 0.732 | **0.582** | **0.692** | **0.67** |
| Perf-only (ours) | 0.600 | 0.695 | **0.657** | **0.721** | 0.664 | **0.740** | 0.527 | 0.648 | 0.66 |
| Melody-only | 0.609 | 0.693 | 0.640 | 0.693 | 0.582 | 0.711 | 0.569 | 0.636 | 0.64 |
| Unconditional | 0.376 | 0.461 | 0.423 | 0.480 | 0.384 | 0.588 | 0.347 | 0.520 | 0.48 |
| **Internal Dataset** | | | | | | | | | |
| Melody & perf (ours) | **0.646** | **0.708** | 0.610 | 0.717 | **0.590** | **0.706** | **0.658** | **0.743** | **0.67** |
| Perf-only (ours) | 0.624 | 0.646 | 0.624 | 0.638 | 0.422 | 0.595 | 0.601 | 0.702 | 0.61 |
| Melody-only | 0.575 | 0.707 | **0.662** | **0.718** | 0.583 | 0.702 | 0.634 | 0.707 | 0.66 |
| Unconditional | 0.476 | 0.580 | 0.541 | 0.594 | 0.400 | 0.585 | 0.522 | 0.623 | 0.54 |

Table 4: Average overlapping area (OA) similarity metrics comparing models with different conditioning. Unconditional and Melody-only baselines are from (Huang et al., 2019b). The metrics are described in detail in Section 4. The samples in this quantitative comparison are used for the listener study shown in the right graph of Figure 4.

the original starting performance, which we call "performance A", and the other half we reserve for the end performance, denoted as "performance B." Then we use the performance encoder to encode performance A into its compressed representation $z_A$, and do the same for performance B to obtain $z_B$. For a range $\alpha \in [0, 0.125, \dots, 0.875, 1.0]$, we sample a new performance $\text{perf}_{\text{new}}$ that results from decoding $\alpha \cdot z_A + (1-\alpha) \cdot z_B$. We observe how the OA (averaged across all features) defined in Section 4 changes between this newly interpolated performance $\text{perf}_{\text{new}}$ and performances $\{A, B\}$.

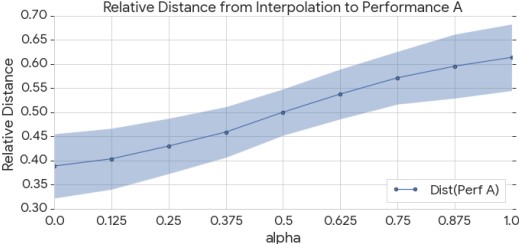

Figure 2: For the internal dataset, the relative distance from performance A ($\alpha = 1$) to the interpolated sample increases as $\alpha$ is slowly increased to 1.0.

Specifically, we compute the similarity metric between each input performance A and interpolated sample $\text{perf}_{\text{new}}$ for all 500 samples, and compute the same pairwise similarity for each performance B. We then compute the normalized distance between each interpolated sample and the corresponding performance A or B, which we denote as: `rel_distance(perf A)` $= 1 - \frac{\text{OA\_A}}{\text{OA\_A + OA\_B}}$, where the OA is averaged across all features. We average this distance across all elements in the set and find in Figure 2 that the relative distance between performance A slowly increases as we increase $\alpha$ from 0 to 1, as expected. We note that it is not possible to conduct this interpolation study with non-aggregated baselines, as we cannot interpolate across variable-length embeddings. We find that a similar trend holds for MAESTRO as in Figure 3(a).

### 5.3.2 MELODY & PERFORMANCE AUTOENCODER

We conduct a similar study as above with the melody & performance autoencoder. We hold out 716 unique melody-performance pairs (melody is not derived from the same performance) from the internal evaluation dataset and 50 examples from MAESTRO. We then interpolate across the different performances, while keeping the conditioning melody input the same across the interpolations.

As shown in Figure 3(a), we find that a similar trend holds as in the performance autoencoder: the newly-interpolated samples show that the relative distance between performance A increases as we increase the corresponding value of $\alpha$. We note that the interpolation effect is slightly lower than that of the previous section, particularly because the interpolated sample is also dependent on the

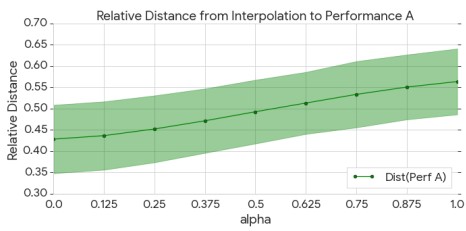 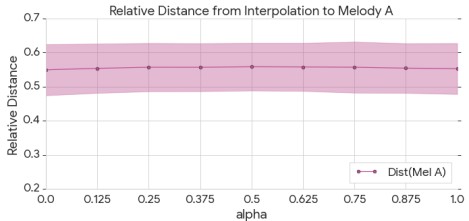

(a) Relative distance to performance A   (b) Relative distance to melody A

Figure 3: For the internal dataset, relative distance from performance A ($\alpha = 1$) as $\alpha$ is slowly increased to 1.0 while the conditioned melody remains fixed. As in (b), we note that the relative distance to the fixed conditioning melody with respect to a random performance remains fixed while the interpolation is conducted between performances A and B, which shows that we can control for elements of style and melody separately.

melody that it is conditioned on. Interestingly, in Figure 3(b), we note that the relative distance between the input performance from which we derived the original melody remains fairly constant across the interpolation procedure. This suggests that we are able to factorize out the two sources of variation and that varying the axis of the input performance keeps the variation in melody constant.

## 5.4 HUMAN EVALUATION

We also conducted listening studies to evaluate the perceived effect of performance and melody conditioning on the generated output. Using models trained on the internal dataset, we conducted two studies: one for performance conditioning, and one for melody and performance conditioning.

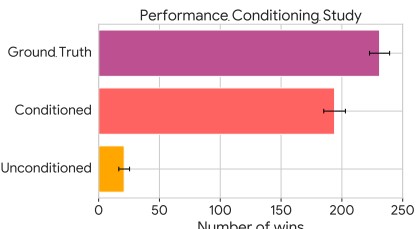 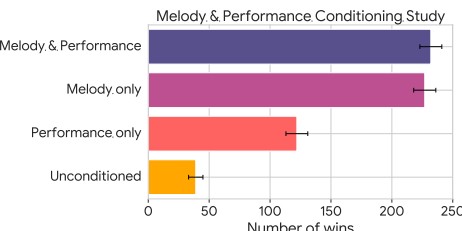

Figure 4: Results of our listening studies, showing the number of times each source won in a pairwise comparison. Black error bars indicate estimated standard deviation of means.

For performance conditioning, we presented participants with a 20s performance clip from the evaluation dataset that we used as a conditioning signal. We then asked them to listen to two additional 20s performance clips and use a Likert scale to rate which one sounded most similar in style to the conditioning signal. The sources the participants rated included "Ground Truth" (a different snippet of the same sample used for the conditioning signal), "Conditioned" (output of the Performance Autoencoder), and "Unconditioned" (output of unconditional model). For this study, 492 ratings were collected, with each source involved in 328 pair-wise comparisons.

For melody and performance conditioning, we presented participants with a 20s performance clip from the evaluation dataset and a 20s melody from a different piece in the evaluation dataset that we used as our conditioning signals. We then asked them to listen to two additional 20s performance clips and use a Likert scale to rate which sounded most like the conditioning melody played in the style of the conditioning performance. The sources the participants rated included "Melody & Performance" (output of the Melody-Performance Autoencoder), "Melody only" (output of a model conditioned only on the melody signal), "Performance only" (output of a model conditioned only on the performance signal), and "Unconditioned" (output of an unconditional model). For this study, 714 ratings were collected, with each source involved in 357 pair-wise comparisons.

Figure 4 shows the number of comparisons in which each source was selected as being most similar in style to the conditioning signal. A Kruskal-Wallis H test of the ratings showed that there is at least

one statistically significant difference between the models: $\chi^2(2) = 332.09$, $p < 0.05$ ($7.72\mathrm{e}{-}73$) for melody conditioning and $\chi^2(2) = 277.74$, $p < 0.05$ ($6.53\mathrm{e}{-}60$) for melody and performance conditioning. A post-hoc analysis using the Wilcoxon signed-rank test with Bonferroni correction showed that there were statistically significant differences between all pairs of the performance study with $p < 0.05/3$ and all pairs of the performance and melody study with $p < 0.05/6$ except between the "Melody only" and "Melody & Performance" models ($p = 0.0894$).

These results demonstrate that the performance conditioning signal has a clear effect on the generated output. In fact, the effect was sufficiently robust that in the 164 comparisons between "Ground Truth" and "Conditioned", participants said they had a preference for "Conditioned" 58 times.

Although the results between "Melody-only" and "Melody & Performance" are close, this study demonstrates that conditioning with both melody and performance outperforms conditioning on performance alone, and they are competitive with melody-only conditioning, despite the model having to deal with the complexity of incorporating both conditioning signals. In fact, we find quantitative evidence that human evaluation is more sensitive to melodic similarity, as the "Performance-only" model performs worst – a slight contrast to the results from the OA metric in Section 5.2.

We provide several audio examples demonstrating the effectiveness of these conditioning signals in the online supplement at `http://bit.ly/2l14pYg`. Our qualitative findings from the audio examples and interpolations, coupled with the quantitative results from the similarity metric and the listening test which capture different aspects of the synthesized performance, support the finding that the Melody & Performance autoencoder offers significant control over the generated samples.

## 6    RELATED WORK

**Sequential autoencoders:** Building on the wealth of autoencoding literature (Hinton & Salakhutdinov, 2006; Salakhutdinov & Hinton, 2009; Vincent et al., 2010), our work bridges the gap between the traditional sequence-to-sequence framework (Sutskever et al., 2014), their recent advances with various attention mechanisms (Vaswani et al., 2017; Shaw et al., 2018; Huang et al., 2019b), and sequential autoencoders. Though (Wang & Wan, 2019) propose a Transformer-based conditional VAE for story generation, the self-attention mechanism is shared between the encoder and decoder. Most similar to our work is that of (Kaiser & Bengio, 2018), which uses a Transformer decoder and a discrete autoencoding function to map an input sequence into a discretized, compressed representation. We note that this approach is complementary to ours, where a similar idea of discretization may be applied to the output of our Transformer encoder. The MusicVAE (Roberts et al., 2018) is a sequential VAE with a hierarchical recurrent decoder, which learns an interpretable latent code for musical sequences that can be used during generation time. This work builds upon (Bowman et al., 2015) that uses recurrence and an autoregressive decoder for text generation. Our Transformer autoencoder can be seen as a deterministic variant of the MusicVAE, with a complex self-attention mechanism based on relative positioning in both the encoder and decoder to capture more expressive features of the data at both the local and global scale.

**Controllable generations using representation learning:** There is also a wide range of recent work on controllable generations, where we focus on the music domain. (Engel et al., 2017a) proposes to constrain the latent space of unconditional generative models to sample with respect to some predefined attributes, whereas we explicitly define our conditioning signal in the data space and learn a global representation of its style during training. The Universal Music Translation network aims to translate music across various styles, but is not directly comparable to our approach as they work with raw audio waveforms (Mor et al., 2018). Both (Meade et al., 2019) and MuseNet (Payne, 2019) generate music based on user preferences, but adopt a slightly different approach: the models are specifically trained with labeled tokens (e.g., composer and instrumentation) as conditioning input, while our Transformer autoencoder's global style representation is learned in an unsupervised way.

## 7    CONCLUSION

We proposed our Transformer autoencoder for conditional music generation, a sequential autoencoder model which utilizes an autoregressive Transformer encoder and decoder for improved modeling of musical sequences with long-term structure. We show that this model allows users to easily

adapt the outputs of their generative model using even a single input performance. Through experiments on the MAESTRO and internal datasets, we demonstrate both quantitatively and qualitatively that our model generates samples that sound similar in style to a variety of conditioning signals relative to baselines. For future work, it would be interesting to explore other training procedures such as variational techniques or few-shot learning approaches to account for situations in which the input signals are from slightly different data distributions than the training set.

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

APPENDIX

## A   FURTHER DETAILS ON INPUT PERTURBATION PROCEDURE

We elaborate upon the perturbation procedure as follows, which *only* applies to the melody & performance conditioning music generation task. First, we take a target performance that we would like our model to predict and extract the corresponding melody from this performance. We use this "clean" melody as part of our conditioning signal. Then, we modify the conditioning performance by either shifting the pitch up or down 6 semitones and stretching the timing by $\pm$ 5%. Then for each new data point during training, a single noise injection procedure is randomly sampled from the cross product of all possible combinations of 12 pitch shift values and 4 time stretch values (evaluated in intervals of 2.5%). At test time, the data points are left unperturbed.

## B   NLL EVALUATION FOR "NOISY" MODEL

Below, we provide the note-wise test NLL on the MAESTRO and internal datasets with melody conditioning, where the conditioning performance is perturbed by the procedure outlined in Section 3.

| Model variation | MAESTRO | internal Dataset |
|---|---|---|
| Noisy Melody TF autoencoder with relative attention, sum | 1.721 | 1.248 |
| Noisy Melody TF autoencoder with relative attention, concat | 1.719 | 1.249 |
| Noisy Melody TF autoencoder with relative attention, tile | 1.728 | 1.253 |

Table 5: Note-wise test NLL on the MAESTRO and internal piano performance datasets with melody conditioning, with event-based representations of lengths $L = 2048$.

## C   ADDITIONAL DETAILS ON MODEL TRAINING PROCEDURE

We emphasize that the Transformer is trained in an *autoencoder-like* fashion. Specifically, for performance-only conditioning, the Transforemr decoder is tasked with predicting the same performance that was fed as input to the encoder. In this way, we encourage the model to learn global representations (the mean-aggregated performance embedding from the encoder) that will faithfully be able to reconstruct the input performance. For melody  performance conditioning, the Transformer autoencoder is trained to predict a new performance using the combined melody+performance embedding, where the loss is computed with respect to the conditioned input performance that is provided to the encoder.

## D   MODEL ARCHITECTURE AND HYPERPARAMETER CONFIGURATIONS

We mostly use the default Transformer architecture as provided in the Tensor2Tensor framework, such as 8 self-attention heads as listed in the main text, and list the slight adjustments we made for each dataset below:

### D.1   MAESTRO

For the MAESTRO dataset, we follow the hyperparameter setup of (Huang et al., 2019b):

1. num hidden layers = 6
2. hidden units = 384
3. filter size = 1024
4. maximum sequence length = 2048
5. maximum relative distance = half the hidden size
6. dropout = 0.1

### D.2 INTERNAL DATASET

For the internal dataset, we modify the number of hidden layers to 8 and slightly increase the level of dropout.

1. num hidden layers = 8
2. hidden units = 384
3. filter size = 1024
4. maximum sequence length = 2048
5. maximum relative distance = half the hidden size
6. dropout = 0.15

## E INTERPOLATIONS DETAILS

Interpolation relative distance results for the (a) performance and (b) melody & performance Transformer autoencoders for the MAESTRO dataset.

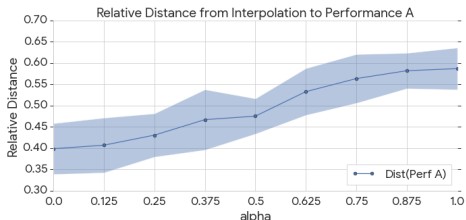 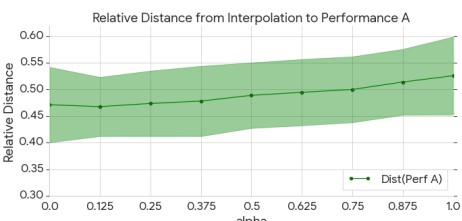

(a) Relative distance from interpolated sample to the original starting performance.

(b) Relative distance from the interpolated sample to the original melody, which is kept fixed.

Figure 5: The distance to the original performance increases as the value of $\alpha$ increases in (a), as expected. In (b), we see that there is a very slight increase in the relative distance to the original melody during the interpolation procedure.

## F    INTERNAL DATASET PERFORMANCE INTERPOLATIONS

Here, we provide piano rolls demonstrating the effects of latent-space interpolation for the internal dataset, for both the (a) performance and (b) melody & performance Transformer autoencoder. For similar results in MAESTRO as well as additional listening samples, we refer the reader to the online supplement: `http://bit.ly/2l14pYg`.

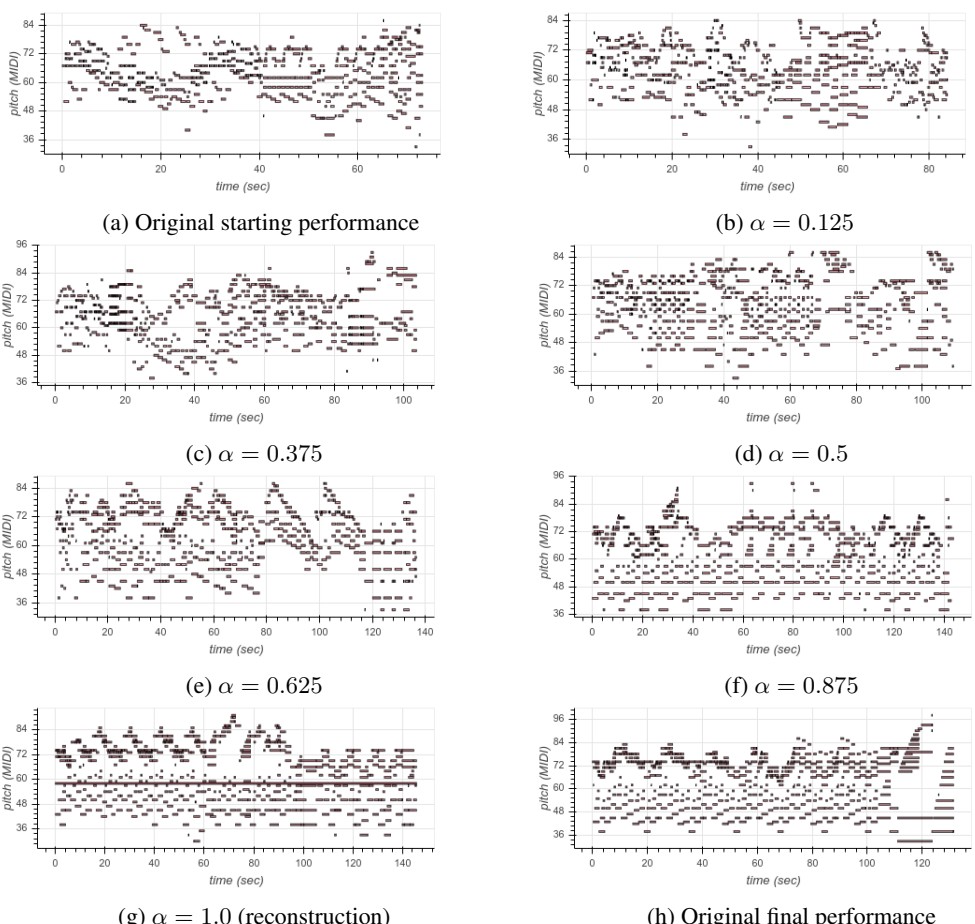

Figure 6: Interpolation of a starting performance (a) from the internal dataset to a final performance (h), with the coefficient $\alpha$ controlling the level of interpolation between the latent encodings between the two performances.

## G    INTERNAL DATASET MELODY INTERPOLATION

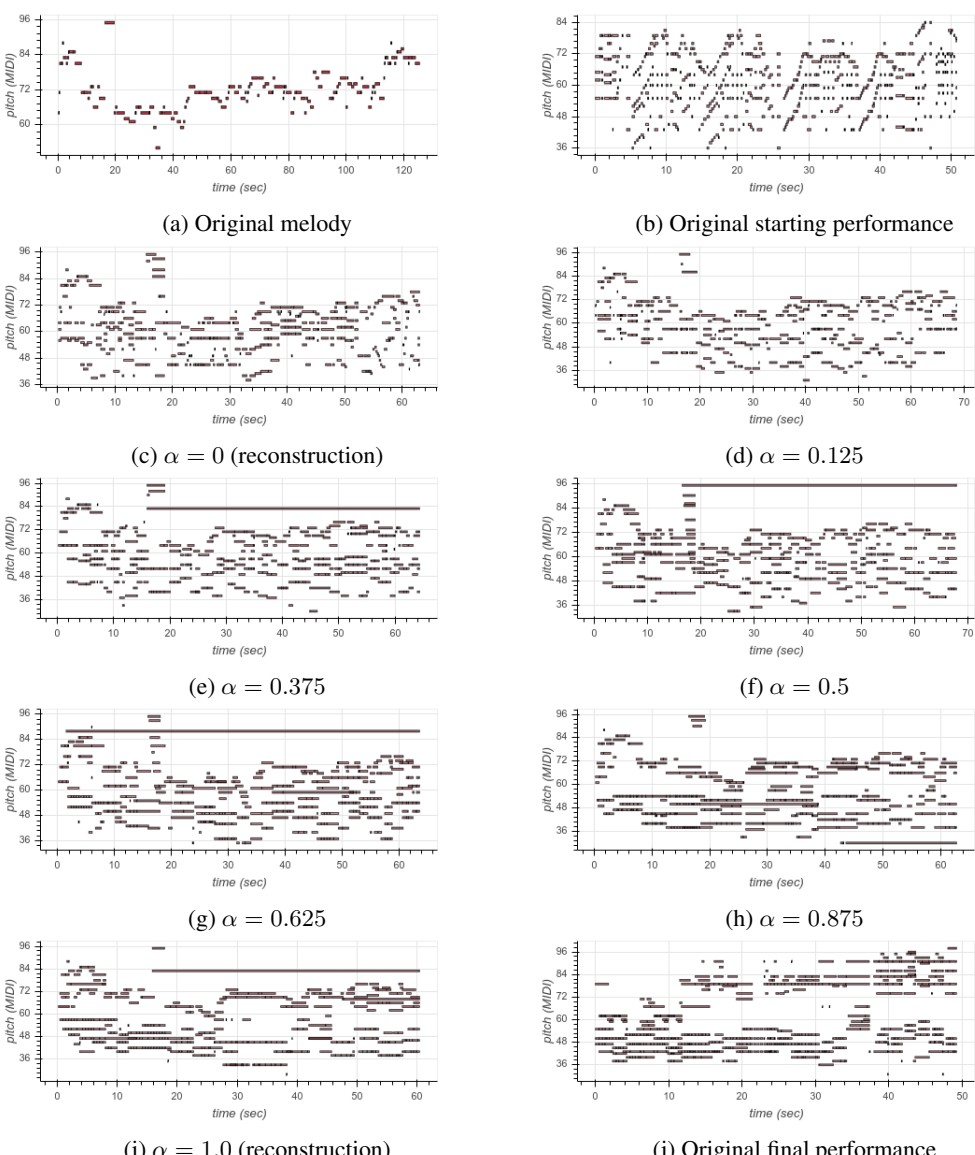

(a) Original melody

(b) Original starting performance

(c) $\alpha = 0$ (reconstruction)

(d) $\alpha = 0.125$

(e) $\alpha = 0.375$

(f) $\alpha = 0.5$

(g) $\alpha = 0.625$

(h) $\alpha = 0.875$

(i) $\alpha = 1.0$ (reconstruction)

(j) Original final performance

Figure 7: Interpolation of a starting performance (b) from the internal dataset to a final performance (j), with the coefficient $\alpha$ controlling the level of interpolation between the latent encodings between the two performances. The original conditioning melody (a) is kept fixed throughout the interpolation.

