# OpenReview forum: "Encoding Musical Style with Transformer Autoencoders"
_ICLR.cc/2020/Conference — Reject_

### Official Review · AnonReviewer3 · 2019-10-23
**Official Blind Review #3**

**Rating:** 3

**Review:**

This paper presents a technique for encoding the high level “style” of pieces of symbolic music. The music is represented as a variant of the MIDI format. The main strategy is to condition a Music Transformer architecture on this global “style embedding”.  Additionally, the Music Transformer model is also conditioned on a combination of both “style” and “melody” embeddings to try and generate music “similar” to the conditioning melody but in the style of the performance embedding.

Overall, I think the paper presents an interesting application and parts of it are well written, however I have concerns with the technical presentation in parts of the paper and some of the methodology. Firstly, I think the algorithmic novelty in the paper is fairly limited. The performance conditioning vector is generated by an additional encoding transformer, compared to the Music Transformer paper (Huang et. al. 2019b). However, the limited algorithmic novelty is not the main concern. The authors also mention an internal dataset of music audio and transcriptions, which can be a major contribution to the music information retrieval (MIR) community. However it is not clear if this dataset will be publicly released or is only for internal experiments.

In terms of technical presentation, I think the authors should clarify how the model is trained. It took me a couple of passes and reading the Music Transformer paper to realise that in the melody and performance conditioning case, the aim is to generate the full score (melody and accompaniment) while conditioning on the performance style and melody (which is represented using a different vocabulary). This point can be easily clarified in Figure 1, by adding the input to the encoder as input to the decoder for computing the loss. Although I understand the need for anonymity and constraints while referring to unreleased datasets, it would still be useful for the reader/reviewer to have some details of how the melody was extracted and represented. “An internal procedure” is quite mysterious.

Measuring music similarity is a difficult problem and the topic has been the subject of at least 2 decades of research. I find the description of the “performance feature” to be lacking in necessary background and detail. Firstly, I am not sure what the final dimensionality of the feature vector is. Is it real valued? The authors mention (Yang and Lerch, 2018) but use a totally different set of attributes compared to that paper. I also don’t see the connection between this proposed feature vector and using the IMQ kernel for measuring similarity. This connection is not motivated adequately and after reading (Jitkrittum et. al. 2019) its not obvious to me why this is the most appropriate metric. Finally, it would be useful if the authors comment on existing methods for measuring music similarity in symbolic music and how their proposed feature fits into existing work. A lot of work has been published on this topic, most recently in the context of Query-by-Humming [1].

Minor Comments

1. “...which typically incorporate global conditioning as part pf the training procedure” Could you elaborate on this point? Is the global conditioning the samples from the noise distribution?
2. Figure 1 should be clarified or another figure should be added to show how the melody conditioning works. Maybe a comment on the melody vocabulary or a reference would also be useful.
3. The MAESTRO dataset is described in terms of the number of performances while the internal dataset is described in terms of the number of hours of audio. Its not possible for the reader to get a sense of the relative sizes of the 2 datasets and how the results should be interpreted.
4. There should be more background and description in Section 4. Where does the performance feature come from? Why use this feature compared to existing techniques for measuring similarity between symbolic music pieces? Is it computational efficiency? Why not compare the conditioning melody with the generated performance similar to query-by-humming? Where does the IMQ kernel come from? What is the size of the feature vector?
5. In section 5.2, a conditioning sample, a generated sequence and an unconditional sample are used to compute the similarity measure. Which terms do these correspond to in the MMD-like term (x,y,y’)?
6. I like the experiments performed in Section 5.3 with the linear combination of 2 performance embeddings.

[1] A Survey of Query-By-Humming Similarity Methods: http://vlm1.uta.edu/~athitsos/publications/kotsifakos_petra2012.pdf


**Experience Assessment:**

I have published one or two papers in this area.

**Review Assessment: Checking Correctness Of Derivations And Theory:**

I assessed the sensibility of the derivations and theory.

**Review Assessment: Checking Correctness Of Experiments:**

I assessed the sensibility of the experiments.

**Review Assessment: Thoroughness In Paper Reading:**

I read the paper thoroughly.

---

> ### Author Response · Authors · 2019-11-08
> **Specific Comments**
>
> In addition to the common concerns as written above, we address Reviewer #3's specific concerns below:
>
> > 1. What does “typically incorporate global conditioning” mean in the Introduction?
>
> The generative models which “typically incorporate global conditioning...” are simply conditional variants of models such as the conditional VAE (Sohn et al. 2015) and conditional GAN (Mizra et. al 2014) which perform generation by conditioning on a global signal, such as a one-hot encoding of the class label.
>
> > 2. Need more clarification about “internal dataset” and “preprocessing procedure” for melody extraction.
> As the reviewer noted, we did our best to anonymize the submission with respect to the dataset and preprocessing techniques used in the paper. Our internal dataset is comprised of approximately 400K piano performances which comprise the 10,000+ hours of audio. Due to licensing restrictions we are unable to release the internal piano performance dataset -- however, we will provide pre-trained models based on this dataset for public use.
>
> For the melody representation (vocabulary), we followed (Waite et. al 2016) to encode the melody as a sequence of tokens and quantized it to a 100ms grid. For the melody extraction procedure, we used an algorithm as in the open-sourced code (Anonymous for review), where we use a heuristic to extract the note with the highest in a given performance. Specifically, we construct a transition matrix of melody pitches and use the Viterbi algorithm to infer the most likely sequence of melody events within a given frame. We will add additional details regarding the melody extraction and encoding in the Supplement.
>
> >3. There needs to be additional clarification of how the model is trained.
>
> This is a good point. As noted by the reviewer, for performance-only conditioning, the decoder is tasked with predicting the same performance that was fed as input to the encoder. In this way, we encourage the model to learn global representations (the mean-aggregated performance embedding from the encoder) that will faithfully be able to reconstruct the input performance. For melody & performance conditioning, the Transformer autoencoder is trained to predict a new performance using the combined melody+performance embedding, where the loss is computed with respect to the conditioned input performance that is provided to the encoder.
>
> To make this point more clear, we will update the submission with a new version of Figure 1 with the reviewer’s suggestions. We have also added these additional details on the model training procedure in the supplemental materials in the revision.
>
>
> References:
> Sohn et. al 2015: Learning Structured Output Representation using Deep Conditional Generative Models
> Mizra et. al 2014: Conditional Generative Adversarial Nets
> Waite et. al 2016: https://magenta.tensorflow.org/2016/07/15/lookback-rnn-attention-rnn

---

> > ### Comment · AnonReviewer3 · 2019-11-15
> > **Thanks for the changes.**
> >
> > Dear Authors,
> >
> > Thank you for all the changes to the draft. I think the paper is much improved due to all the changes. I need some time to go through all the changes in detail and reconsider my rating for the paper.

---

### Official Review · AnonReviewer1 · 2019-10-23
**Official Blind Review #1**

**Rating:** 6

**Review:**

## summary
In this paper, the author extends the standard music Transformer into a conditional version: two encoders are evolved, one for encoding the performance and the other is used for encoding the melody. The output representation has to be similar to the input. The authors conduct experiments on the MAESTRO dataset and an internal, 10,000+ hour dataset of piano performances to verify the proposed algorithm.

## Novelty
The application is interesting, but the novelty of the architecture itself is limited. Multiple encoder structure has been widely investigated in machine translation.

## Questions
1.	In section 4.2, how do you use the $\mathcal{Y}$? Since it is defined but never used. What does the $p()$ and $q()$ mean ? You mentioned that “We omit the usual first term in the MMD loss …” but if so, why do you introduce this term to evaluation metric?
2.	By checking the music Transformer, in Table 3, it is not surprising to see that the proposed method outperforms the corresponding baselines, because no conditional information is used.
3.     It is better to give some mathematical definition of music generation with specific style. I am not working on music generation but I list two CV related papers about conditional image translation, which mathematically describes "an image with specific style".
4.	Considering that this is an unsupervised setting that two styles are transformed, can cycle-consistency be implemented as a baseline? The following two papers are about conditional unsupervised image-to-image translation, which build a cycle-consistency loss during the feedback and might help improve the performances.


## Reference
[ref1] Multimodal Unsupervised Image-to-Image Translation, ECCV’18
[ref2] Conditional image-to-image translation, CVPR’18


**Experience Assessment:**

I do not know much about this area.

**Review Assessment: Checking Correctness Of Derivations And Theory:**

I assessed the sensibility of the derivations and theory.

**Review Assessment: Checking Correctness Of Experiments:**

I assessed the sensibility of the experiments.

**Review Assessment: Thoroughness In Paper Reading:**

I read the paper at least twice and used my best judgement in assessing the paper.

---

> ### Author Response · Authors · 2019-11-08
> **Specific comments**
>
> In addition to the common concerns as written above, we address Reviewer #1's specific concerns below:
>
> 1. Is there a mathematical definition of style-specific generation, with more relevant baselines (e.g. cycle-consistency)?
> We appreciate the reviewer’s suggestion and additional references. Because our method is learning a conditional generative model, we do not incorporate any additional style-specific terms in our learning objective as we perform maximum-likelihood training. However, developing a more fine-grained notion of style for music generation is certainly interesting.
>
> We originally compared against the Music Transformer to ensure that conditioning would improve the model, and added various versions of our model (e.g. melody-only conditioning) as relevant baselines for comparison. To the best of our knowledge, our work is the first to incorporate conditioning information in sequential language models for music generation.
>
> Adding a consistency loss term is a good idea for a baseline, but for our problem it was ill-posed because we did not have a straightforward way of partitioning our data into different categories. In image translation literature (e.g. Zhu et. al 2018), there exists a clear source and target domain even if the images themselves are unlabeled. For both MAESTRO and the internal dataset, such separations between the source and target domains are unclear (e.g. musical tempo, rhythm, pitch, etc.). Nevertheless, we agree that this would be interesting to explore as future work.
>
> References:
> Zhu et. al 2018: Unpaired Image-to-Image Translation using Cycle-Consistent Adversarial Networks

---

### Author Response · Authors · 2019-11-08
**Addressing Common Concerns**

We thank the reviewers for their insightful comments! Before we address common threads of concern below, then respond to each reviewer individually, we highlight significant changes we have made to the paper:

- Following helpful reviewer suggestions, we have completely reworked the evaluation section (Section 4) of the paper, and uploaded a *revised* version of the paper.
- For more meaningful and fine-grained evaluation, we have adopted a collection 8 commonly used musical similarity metrics with which to compare samples.
- As an aggregate similarity metric, we now use the average of the 8 similarity metrics, rather than the IMQ kernel, which is more intuitive and better motivated.
- We have added tables 3 and 4 to report these fine-grained new metrics, and updated figures 2 and 3 to use the new aggregate similarity metric (Section 5).
- All of the key findings of the paper remain the same with these new metrics, and many effects are actually more pronounced than with the original kernel similarity metric.

> R1/R3: Why not use existing techniques for measuring similarity between musical performances? Why not compare the conditioning melody with the generated performance similar to query-by-humming (QBH)?

We appreciate the reviewers’ feedback regarding our kernel evaluation metric. Upon further reflection, we agree that there are simpler and more intuitive ways to evaluate musical similarity, and as described above we have reworked large parts of the paper to reflect that. In the revised version of the paper, we follow existing techniques (Yang & Lerch 2018) using the Overlapping Area (OA) of common similarity metrics including:
- Note Density
- Pitch Range
- Mean/Var Pitch
- Mean/Var Velocity
- Mean/Var Duration
We note that certain features in the paper are not applicable to our setting (e.g. note length transition matrix) because they were developed for monophonic melodies, while we are evaluating polyphonic piano performances.

Although we no longer use the IMQ kernel as the similarity metric, we emphasize that the key results remain the same. For clarity, we quickly examine here why that is the case and what originally motivated the kernel approach. The kernel approach assumes that the conditioning performances (x~p(x)) and generated performances (y, y’~q(y)) are drawn from two different distributions, and MMD computes the degree to which these two distributions are similar in a kernel feature space. We experimented with a variety of kernels commonly used in the literature (e.g. RBF kernel) and found that the IMQ worked best empirically. Our results remained unchanged because the MMD distributional similarity correlates well with the average difference in extracted similarity features.


We did not compare an input melody to the generated performance (as in QBH) because we wanted to compare the similarities of polyphonic sequences. As the melody is represented using a different encoding and vocabulary than the performance, comparison is not straightforward and would not provide relevant information. We do note that our user listening studies also implicitly serve as a proxy to measure melodic similarity. Thus we performed our similarity evaluations against the original performance from which the melody was extracted. This is reflected in the melody & performance conditioning case: we average two OA terms, OA(source performance of extracted melody, generated sample)  and OA(conditioning performance, generated sample), as our final metric. In this way, we account for the contributions of both the conditioning melody and performance sequence.

> R1/R3: Algorithmic novelty is somewhat limited.
We emphasize that our goal is to provide users with more fine-grained control over the outputs generated by a seq2seq language model. Despite its simplicity, our method is able to learn a global representation of style for a Transformer, which to the best of our knowledge is a novel contribution for music generation. Additionally, we can synthesize an arbitrary melody into the style of another performance, and we demonstrate the effectiveness of our results both quantitatively (metrics) and qualitatively (interpolations, samples, and user listening studies).

References:
Hung et. al 2018: Improving Automatic Jazz Melody Generation by Transfer Learning Techniques
Yang & Lerch 2018: On the evaluation of generative models in music

---

### Decision · Program_Chairs · 2019-12-19

**Decision:**

Reject

**Comment:**

Main content:

Blind review #3 summarizes it well:

This paper presents a technique for encoding the high level “style” of pieces of symbolic music. The music is represented as a variant of the MIDI format. The main strategy is to condition a Music Transformer architecture on this global “style embedding”.  Additionally, the Music Transformer model is also conditioned on a combination of both “style” and “melody” embeddings to try and generate music “similar” to the conditioning melody but in the style of the performance embedding.

--

Discussion:

The reviewers questioned the novelty. Blind review #2 wrote: "Overall, I think the paper presents an interesting application and parts of it are well written, however I have concerns with the technical presentation in parts of the paper and some of the methodology. Firstly, I think the algorithmic novelty in the paper is fairly limited. The performance conditioning vector is generated by an additional encoding transformer, compared to the Music Transformer paper (Huang et. al. 2019b). However, the limited algorithmic novelty is not the main concern. The authors also mention an internal dataset of music audio and transcriptions, which can be a major contribution to the music information retrieval (MIR) community. However it is not clear if this dataset will be publicly released or is only for internal experiments."

However, after revision, the same reviewer has upgraded the review to a weak accept, as the authors wrote "We emphasize that our goal is to provide users with more fine-grained control over the outputs generated by a seq2seq language model. Despite its simplicity, our method is able to learn a global representation of style for a Transformer, which to the best of our knowledge is a novel contribution for music generation. Additionally, we can synthesize an arbitrary melody into the style of another performance, and we demonstrate the effectiveness of our results both quantitatively (metrics) and qualitatively (interpolations, samples, and user listening studies)."

--

Recommendation and justification:

This paper is borderline for the reasons above, and due to the large number of strong papers, is not accepted at this time. As one comment, this work might actually be more suitable for a more specialized conference like ISMIR, as its novel contribution is more to music applications than to fundamental machine learning approaches.